# Biocompatibility Assessment of Two Commercial Bone Xenografts by In Vitro and In Vivo Methods

**DOI:** 10.3390/polym14132672

**Published:** 2022-06-30

**Authors:** Carlos Humberto Valencia-Llano, Diego López-Tenorio, Carlos David Grande-Tovar

**Affiliations:** 1Grupo Biomateriales Dentales, Escuela de Odontología, Universidad del Valle, Calle 4B # 36-00, Cali 76001, Colombia; carlos.humberto.valencia@correounivalle.edu.co (C.H.V.-L.); dilote@hotmail.com (D.L.-T.); 2Grupo de Investigación de Fotoquímica y Fotobiología, Universidad del Atlántico, Carrera 30 Número 8-49, Puerto Colombia 081008, Colombia

**Keywords:** biocompatibility, bone substitutes, collagen, xenografts

## Abstract

Bone substitutes based on xenografts have been used for a long time in bone regeneration thanks to their inductive capacity for bone tissue regeneration. Some bone-based scaffolds have been modified by adding collagen and other proteins to improve their regenerative capacity and prevent migration and aggregation, especially particles. However, rejection of this graft has been reported due to protein residues caused by poor material preparation. We compared the in vitro and in vivo biological response of two commercial xenografts (InterOss^®^, F1 and InterOss^®^ Collagen, F2) and a commercial porcine collagen membrane (InterCollagen^®^ Guide, F3) as a rapid degradation control. Fourier Transform Infrared Spectroscopy (FT-IR) analysis evidenced the presence of hydroxyl, orthophosphate, and carbonate groups of the xenografts and amide groups of collagen. Thermogravimetric analysis (TGA) of the xenografts demonstrated their thermal stability and the presence of a few amounts of organic material. The study by differential scanning calorimetry showed the presence of endothermic peaks typical of the dehydration of the xenografts (F1 and F2) and for the collagen membrane (F3), the beginning of structural three-dimensional protein changes. Subsequently, in vitro biocompatibility tests were carried out for the materials with *Artemia salina* and MTT cell viability with HeLa cells, demonstrating the high biocompatibility of the materials. Finally, in vivo biocompatibility was studied by implanting xenografts in biomodels (Wistar rats) at different periods (30, 60, and 90 days). The F1 xenograft (InterOss) remained remarkably stable throughout the experiment (90 days). F2 (InterOss Collagen) presented a separation of its apatite and collagen components at 60 days and advanced resorption at 90 days of implantation. Finally, the collagen membrane (F3) presented faster resorption since, at 90 days, only some tiny fragments of the material were evident. All the in vivo and in vitro test results demonstrated the biocompatibility of the xenografts, demonstrating the potential of these materials for tissue engineering.

## 1. Introduction

Biocompatibility is a quality that materials that are going to be in contact with living tissues must present, and is defined as the ability of the material to interact with cells in a living environment without causing damage or encouraging an adverse reaction [1]. Biocompatibility is directly related to cytotoxicity, or the material’s ability to produce damage to a biological system [2]. 

To stimulate the regeneration of bone tissue, two types of materials are available: grafts and bone substitutes. Both can promote bone regeneration, but they differ because the graft is living tissue while the substitutes are only mineralized bone matrix without cells [3]. Due to their origin, the materials with application in regenerative techniques can be considered autologous when they come from the same patient, homologous or allogeneic when they originate in individuals of the same species, and xenogeneic when they come from a different species. Finally, alloplastic materials are obtained directly in a laboratory [4].

Alloplastic materials have been developed and are proposed as potential bone substitutes. Their outstanding versatility allows them to be processed with different techniques, such as 3D printing, to obtain porous structures that mimic cancellous bone [5] or resemble the fibrillar portion of the extracellular matrix through nanofibers [6].

Synthetic materials also possess the advantageous possibility of incorporating components with anti-inflammation and antibiotic effects to reduce the possibility of complications [7] or biological factors, such as bone morphogenic proteins (BMP-9), to achieve an osteoinductive effect [8].

Several bone substitutes currently use synthetic materials combined with natural products modified in the laboratory, such as collagen, to achieve biomimetic scaffolds that resemble the extracellular bone matrix [9].

Mammalian bones, fish bones and scales, bird eggshells, and exoskeletons of marine organisms have been used to prepare xenografts (xenogeneic materials) [10]. These products are biocompatible and biodegradable since their structure and chemical composition are similar to human bone tissue [11,12].

Xenograft-type bone substitutes have been used for more than 30 years in different applications of bone regeneration, finding an osteoconductive effect [12,13]. However, there are reports of graft rejection attributed to inadequate preparation of the material that allowed the presence of residual proteins [14].

Different products of xenogeneic origin supplemented with collagen are currently available to improve its biological activity in enhancing regenerative properties. Collagen incorporation into xenografts occurs since collagen is the main extracellular matrix component and to which osteopromoting properties are attributed, such as the stimulation of cell differentiation in osteoblasts and the stimulation of cicatrization [13,15,16]. 

It has been proposed that incorporating collagen into the mineralized bone matrix could improve the behavior of the substitute by adding an osteopromoting quality, allowing improvements in the preservation of bone levels after regenerative procedures [15]. Furthermore, it has been reported that adding collagen membranes to grafted bone substitutes decreases the risk of migration and aggregation [16]. However, studies are needed to assess how adding collagen improves the biocompatibility and stability of the product without affecting creeping substitution, a necessary condition for its success as a bone regenerating material.

This work studies the biological behavior of xenografts incorporated with collagen in vitro and in vivo to demonstrate the materials’ biocompatibility with cells and within subdermal tissues of biomodels. 

## 2. Materials and Methods

### 2.1. Materials

Table 1 shows the formulations of this investigation’s three commercially available materials. These materials are xenografts used in bone regeneration procedures: InterOss (SigmaGraft, Inc., Fullerton, CA, USA) is an organic cancellous bone graft granules size 250–1000 µm [17]; and InterOss Collagen (SigmaGraft, Inc., Fullerton, CA, USA) is an anorganic composite of hydroxyapatite and collagen, composed of a 90 wt.% of bovine bone particles (250–1000 µm in diameter) and a 10 wt.% of collagen fibers of porcine origin. Finally, according to the manufacturer, a porcine collagen membrane (InterCollagen Guide, SigmaGraft, Inc., Fullerton, CA, USA) with applications in periodontics and surgery was used as a positive control for rapid biosorption.

### 2.2. Characterization of the Xenografts

Fourier Transform Infrared Spectroscopy (FT-IR)

Functional group analysis used the Fourier transform infrared spectroscopy in an IR affinity-1 infrared spectrophotometer (Shimadzu, Kyoto, Japan) at the wavenumbers between 500–4000 cm^−1^ in transmittance mode.

#### Thermal Analysis

We conducted the thermal gravimetric analysis using a TA Instrument TGA Q50 V20.13 Build 39 (TA instrument, New Castle, DE, USA). The degradation temperature range was 30 and 800 ± 2 °C under nitrogen at a flow rate of 60 mL/min and a heating rate of 10 °C/min. 

The fusion (T_m_) temperatures were measured with a DSC2A-00181 system (TA Instruments) with a heating/cooling rate of 5 °C/min from −25 °C/250 °C/−25 °C using the differential scanning calorimetric (DSC) technique. TGA and DSC result analysis was performed using the TA Instruments’ Universal Analysis software

### 2.3. Preliminary Biocompatibility Study with Artemia salina

For the biocompatibility tests with *Artemia salina*, unhatched cysts were used. To obtain the larvae, the following protocol was followed: one gram of *Artemia salina* cysts (brine shrimp eggs, Brinesh-rimpdirect, Ogden, UT, USA) was hydrated in two liters of artificial seawater solution (25 g NaCl/L distilled water), with aeration and permanent lighting [18]. 

After 48 h of hydration, the hatched nauplii were recovered and placed in a box of 24 wells with artificial seawater solution. The materials (F1, F2, F3) were introduced in each well with ten larvae. As a positive control, 3 mL of an enzymatic detergent (Bonzyme, Eufar Laboratories, Bogotá, Colombia) was used, and a seawater solution with the larvae was used as a negative control. All experiments were performed in triplicate. Checks were made at 24 and 48 h of incubation, and the number of live larvae in each well was counted. With that number, the percentage of cytotoxicity was determined with the formula [19]:(1)$$\mathrm{Mortality}\;(\%)=\frac{\mathrm{Number}\;\mathrm{of}\;\mathrm{AS}\;\mathrm{death}\;}{\mathrm{Number}\;\mathrm{of}\;\mathrm{AS}\;\mathrm{Initial}}\;\times \;100$$

### 2.4. Cell Viability Assay

Preliminary cytotoxicity studies were performed using the salt (3-(4,5-dimethylthiazol-2-yl)-2,5-diphenyltetrazolium bromide) MTT, with the cell viability kit MTT cell proliferation assay kit (Abcam ab211091).

Initially, cells of the HeLa-ATCC line available in the in vitro cell culture laboratory of the Universidad del Valle (Cali, Colombia) were thawed and cultured in an incubator with 5% CO_2_. Subsequently, they were transferred to T25 flasks until confluence was obtained. Cells were then detached by trypsinization, and 30,000 cells per well were deposited in the 96-well box under a 5% CO_2_ humidified atmosphere at 37 °C for 24 h.

An enzymatic soap (Bonzyme, Eufar Laboratories, Bogotá, Colombia) was used as a positive control, and cells with a culture medium were used as a negative control. Subsequently, the culture medium was removed and washed with Hank’s balanced salt solution (HBSS) (Gibco, Thermo Fisher Scientific, Waltham, MA, USA). The samples of the materials to be studied were deposited, and HBSS was added to incubate for 24 h.

We followed the protocol recommended in the product’s technical datasheet to conduct the cytotoxicity test. Initially, the culture medium and samples were removed from each well and washed with PBS. Subsequently, 50 µL of HBSS medium and 50 µL of MTT reagent were added to each well. The mixture was incubated for three hours at 37 °C (New Brunswick Galaxi 48R CO_2_ Incubator, Eppendorf Company, Hamburg, Germany). Finally, 150 µL of the solvent in the MTT kit was added, protected from light with a dark field, and shaken on an orbital shaker (Stuart orbital shaker, Fisher Scientific, Waltham, MA, USA) for 15 min. OD measurements were made at 590 nanometers. For the calculation of cytotoxicity, Equation (2) was used to calculate the percentage of MTT viability [20]:Cell viability MTT (%) = Absorbance sample/Absorbance untreated × 100(2)

### 2.5. Cell Growth Assay

For this assay, mesenchymal bone cells were obtained from the cranial bone of five-day-old Wistar rats, according to the protocol proposed by Bakker & Klein-Nulend [21]. 

After three weeks of culture, the bone cells were transferred to T25 flasks with Gibco culture medium (Gibco Gibco, Thermo Fisher Scientific, Waltham, MA, USA), which reached confluence in two weeks. Upon confluence, cells were detached using trypsin (Gibco Gibco, Thermo Fisher Scientific, Waltham, MA, USA). Next, 30,000 cells per well were seeded in a 24-well box with a complete culture medium. The samples (F1, F2, and F3) were deposited for six days in a CO_2_ incubator (New Brunswick Galaxi 48R, Eppendorf Company, Hamburg, Germany). The culture medium was renewed every three days to avoid nutrient depletion. Photomicrographs were acquired with a camera attached to an inverted field microscope (ZEISS, Stuttgart, Germany).

The cultures were performed in triplicate, and three wells with cells in a culture medium without the samples were used as control.

### 2.6. In Vivo Biocompatibility Study

Subdermal implantations were performed on nine male biomodels (Wistar rats) which were five months old and had an average weight of 370 g. The biomodels were distributed in three groups to make observations at 30, 60, and 90 days of implantation.

Ketamine 70 mg/kg (Blaskov Laboratory, Bogotá, Colombia) and Xylazine 30 mg/kg (ERMA Laboratories, Celta, Colombia) were used as anesthetic medication. Once the appropriate level of sedation was obtained, trichotomy of the dorsal surface of the biomodels was performed. To do this, it was disinfected with isodine solution (Laboratorio Sanfer, Bogotá, Colombia) and with infiltrative anesthesia (Lidocaine 2%) with epinephrine (Newstetic, Guarne, Colombia). Subsequently, subdermal preparations of one centimeter wide by five centimeters deep were prepared.

After implantations, the incision was closed with a four-zero silk suture. In addition, Gentamicin 0.1 wt.% was applied on the wound (Procaps, Barranquilla, Colombia) and Diclofenac 75 mg was applied intramuscularly (La sante, Bogotá, Colombia).

Once the implantation period was over (30, 60, or 90 days), the biomodels were euthanized by intraperitoneal injection of sodium pentobarbital/sodium diphenylhydantoin at a dose of 0.3 mL/kg of weight (Euthanex, INVET Laboratory, Cota, Colombia). The recovered samples were fixed in buffered formalin for 48 h. Following, we washed samples with PBS and processing with Autotechnicon Tissue ProcessorTM equipment (Leica Microsystems, Mannheim-Germany). Next, they were embedded in paraffin blocks (Thermo ScientificTM HistoplastTM, Thermo Fisher Scientific, MA, USA). These samples were cut to six microns with a Leica RM2125 RTS microtome (Leica Microsystem, Mannheim, Germany). For histological studies, a staining sample was performed with the Hematoxylin-Eosin (H-E) technique and Masson’s trichrome (MT). Microphotographs were taken with a Leica DM750 optical microscope and a Leica DFC 295 camera. Images were processed with Leica Application Suite version 4.12.0 software (Leica Microsystem, Mannheim-Germany).

The ethics committee endorsed this research with experimental animals of the Universidad del Valle (Cali, Colombia) through resolution No. 003 of 2020.

### 2.7. Statistic Analysis

Absolute and relative frequencies were calculated for mortality from *Artemia salina* (AS) at 24 h, 48 h, and total mortality. Mortality ratios were contrasted by chi-square distribution to determine the differences between the materials and the negative control. The number of deaths at 24 and 48 h was averaged and contrasted using the Kruskal Wallis and Dunn’s paired tests. The significance level was set at 0.05. Statistical differences for the MTT test were evaluated using Prism 9 version 9.1.1 (GraphPad Software), which calculated the measures of central tendency, dispersion, and position. Average contrast analysis was performed between the first and second tests for each formulation with the Mann–Whitney test. The distributions of absorbance and viability between materials were contrasted using the Kruskal Wallis test and Dunn’s test between pairs. The level of significance was established at 0.05. 

## 3. Results and Discussion

This research studied the biological behavior of three materials applicable in bone tissue engineering using in vitro experiments and in vivo implantations. The first sample is a particulate inorganic bovine apatite (F1, InterOss). According to the manufacturer, it is very stable in the long term, highly porous, resorbable, and representative of a mineralized osteoconductive structure that works well as a substitute for human bone [17]. This type of bone graft is obtained by a two-stage chemical method that generates a cancellous bone graft (with mesoporosity, microporosity, and macroporosity), widely used as a bone grafting substitute due to its immune response and its osteoinductive and osteoconductive characteristics. Additionally, this material has a large surface area (88.2 ± 0.015 m^2^ g^−1^) and a chemical composition rich in Ca (63 wt.%) and P (33 wt.%) in addition to shallow protein content (0.04 ± 0.0015%). The surface area of this material is within the range of the surface area of human bone (50–100 m^2^ g^−1^), facilitating its resorption and application as a bone substitute. The high degree of porosity, the chemical composition, and the large surface area guarantee osteoconduction and growth of bone tissue within the pores [22]. 

On the other hand, a composite scaffold of collagen and inorganic particles of bovine apatite (F2, InterOss Collagen) with high porosity promotes mineralized angiogenesis. This composite results in improved activity for applications in tissue regeneration. It solves the problem that particulate bone fillers (such as F1) typically present with the possibility of collapse and aggregation and that of membranes with little resistance, causing discomfort to the patient and risk of infection [16]. It has been previously shown that this composite scaffold has a cuboid morphology with an irregular distribution of its macro and microporosity [23]. Furthermore, in that study, InterOss Collagen composite (F2) was shown to have a large surface area (77.0 ± 0.2 m^2^ g^−1^), again in the range of the surface area of human bone.

Finally, we have a porcine collagen membrane (F3, InterCollagen Guide) with high porosity, fluid absorption, and resorption capacity, stretchable in all directions with fibrous characteristics. However, this kind of membrane has the drawback of fast degradation and the risk of infections. Due to the scarce information available, we analyze the preliminary biocompatibility of these materials at the in vitro and in vivo levels in subdermal implantations. 

### 3.1. Xenograft Characterization

#### 3.1.1. Fourier-Transform Infrared Spectroscopy (FT-IR)

Figure 1 represents the FT-IR of each xenograft evaluated in this investigation. For F1 (black line), a 3741 band corresponding to the OH^−^ ion stretching vibration in the HA lattice is observed [24]. The bands at 3600 and 1680 cm^−^^1^ from hydroxyapatite (HA) and water hydroxyl (OH^−^) stretching vibrations are also observed in the spectrum [25]. 

Additionally, the characteristic peaks of the CO_3_^−2^ groups of the hydroxyapatite mineralized material are observed at 1010 cm^−1^, 1462 cm^−1^, and 1510 cm^−1^ [17]. On the other hand, a characteristic peak for orthophosphates (PO_4_^−3^) is observed at 1130 cm^−1^. The band at 2360 cm^−1^ can be attributed to carbon dioxide adsorption in the anorganic bone since carbonates are suitable solvents for atmospheric carbon dioxide [26,27]. 

On the other hand, for F2 (red line), the characteristic bands of bovine hydroxyapatite from InterOss particles (90 wt.%) in the composite and some typical bands of collagen in a smaller proportion (10 wt.%). At 3370 cm^−1^ and 1636 cm^−1^, the stretching vibration characteristic bands of hydroxyl groups (OH^−^) are observed. The 874 cm^−1^, 961 cm^−1,^ and 1412 cm^−1^ peaks correspond to the bovine hydroxyapatites (HA) CO_3_^−2^ groups. The orthophosphate peak at 1025 cm^−1^ (with a small shoulder at 1130 cm^−1^) is also observed. The bands at 1650 cm^−1^ and 1550 cm^−1^ corresponding to amide I and amide II of collagen might overlap with the CO_3_^−2^ bands and hydroxyl groups bands due to the high ratio (9:1) between HA bovine particles and collagen [23]. Finally, for F3, the typical bands of purified collagen were observed. A band at 1700 cm^−1^ with a small shoulder at 1637 cm^−1^ was assigned to the C=O group of amide I. We attributed the band at 1548 cm^−1^ to the N-H and C-N groups in amides II. The bands at 3420–3318, 3078, and 2942 cm^−1^ correspond to stretching vibrations of the N-H group in the amide I and the symmetric tensions of CH_2_, respectively [28]. The peaks at 1210 and 1236 cm^−1^ correspond to amide III bands from N-H bending, C-N, and C-H stretching [29]. The band at 1447 cm^−1^ is attributed to a strong C=H stretching from hydrogen bonding in the collagen structure [30]. The peaks at 1031 cm^−1^ and 1400 cm^−1^ might indicate the material’s presence of ester and residual carboxylate groups [31]. 

#### 3.1.2. Thermal Characterization

##### Thermal Gravimetric Analysis (TGA)

Figure 2 shows the thermal degradation patterns of the xenografts (F1 and F2) and the collagen membrane (F3). F1 and F2 (black and red lines, respectively), due to their content of inorganic material (HA), present a plateau region (due to the low content of organic matter between 280–650 °C), something previously observed for this kind of xenografts [17]. Due to the collagen content in F2 (10 wt.%), weight losses at 327 °C (Figure 2B) and 766 °C (Figure 2B) are observed probably due to the removal of CO_2_. Clearly, at 800 °C, there is a higher mass content for F1 due to the higher inorganic content than in F2. The residual content of the material corresponds to the inorganic content of the anorganic bones [17]. Both F1 and F2 presented a slight initial loss of water (3.0–5.0%) with peaks centered at 29 °C (Figure 2B), the content being higher for F1 due to the greater abundance of hydroxyl groups and water. On the other hand, the thermal degradation of F3, the collagen membrane shows a different degradation profile, in three different stages (first stage between 40–200 °C, second stage between 200–450 °C, and third stage between 450–800 °C). In the first stage, a peak centered at 42 °C is observed, corresponding to a loss of 12 wt.% of water. In the second stage, there is a maximum degradation temperature (T_max_) at 334 °C (Figure 2B) corresponding to the loss of a 64 wt.% due to the combustion of collagen and the pyrolysis of its bonds [32]. Finally, we observed in the third stage of degradation a loss of 5 wt.% corresponding to inorganic calcium residues [32].

##### Differential Scanning Calorimetric (DSC) 

Using the differential scanning calorimetry (DSC) technique, the heat flow and thermal transitions of the anorganic bone particles F1 (InterOss), the anorganic bone and porcine collagen scaffold F2 (InterOss Collagen), and the membrane porcine collagen F3 (InterCollagen Guide) (Figure 3) were measured. The DSC curve for F1 shows a small endothermic peak due to dehydration and loss of hydroxyl groups (OH^−^) at 73 °C. In the case of F2, a broader endothermic peak centered at 82 °C is observed. Loss of hydrogen bonds between hydroxyl groups of the bovine HA of InterOss and the collagen is the leading cause. In addition, collagen undergoes a transition due to heating its triple-helical helix structure to a random coil conformation due to dehydration and breaking hydrogen bonds and free water [23,33]. For F2, a second peak centered at 226 °C is also observed, which corresponds to the conformational changes that the helical structure undergoes, the loss of chemically bound water that accompanies the degradation of small compounds, and the start of denaturation protein (loss of tertiary structure) [23,32]. In the F3 curve, only an endothermic peak centered at 76 °C corresponding to the loss of surface water from the membrane is observed. It has been observed that more complex degradation processes, such as combustion, commonly occur for collagen above 300 °C, but the present analysis did not observe this [32]. However, from the TGA analysis, it was possible to evidence this degradation as a product of the combustion around 334 °C (Figure 2A,B).

### 3.2. Biocompatibility Assay with Artemia salina

The *A. salina* test involves exposing 24-h hatched larvae (instars II-III) to different materials or concentrations of the same material to determine the material’s biocompatibility or the lethal dose 50 (LC_50_) of a specific concentration of the material [34]. Larvae are expected to be affected by the material if it is not compatible. This toxicity test is considered reliable and replicable [35]. 

The *A. salina* test for biomaterials was proposed in 1956 [36], but its standardization was carried out in the early 1980s by the reference center for *Artemia* (ARC-Test). Initially, it was developed as a test to study acute toxicity in biomaterials [37]. Since then, numerous studies have been carried out by different scientific disciplines for being a simple, economical, reliable, sensitive, and reproducible method [34,38,39,40]. The *Artemia salina* test is considered highly reliable and sensitive for studying cytotoxicity in materials [34,40,41,42].

*Artemia* larvae can be affected by the product’s toxic components or by the effect of ingested particles on the intestine [43,44,45]. Figure 4 corresponds to the appearance of *Artemia* cysts before and after being hydrated for 24 h.

In this research, F1 (InterOss) and F2 (InterOss Collagen) have a high content of particulate material. For that reason, they can be ingested if their size is small enough or if their dilution due to the effect of seawater produces residues smaller than 50 µm [42]. In the case of the materials with collagen fibers, since they cannot be ingested, high mortality will be related directly to a direct toxic effect. 

Figure 5 corresponds to the samples after 24 h of incubation with *A. salina*. For the experiment to be considered successful, the mortality rate in the negative control group must be less than 10% after 24-h culture [31]. For F1, F2, and F3, the product-inducing mortality more significant than 30% will be considered toxic [32].

The first observation of the wells at 24 h showed that most of the larvae were alive, with zero deaths in the negative control group and the group with the F3 sample (InterCollagen Guide). In contrast, at 48 h, mortality had increased for all groups, being higher in F2 (InterOss Collagen). Table 2 shows the percentage of mortality of larvae in the three wells corresponding to our samples. The three formulations had low mortality rates at 48 h of observation. 

The products’ results showed a high degree of biocompatibility at 24 h, which decreased slightly at 48 h (Table 2). Additionally, the results demonstrated that there were no significant differences between treatments and treatments with the negative control; but, there was a significant difference (*p* < 0.01) between treatment F3 with the positive control, indicating the high biocompatibility of the materials (Figure 6). We expected a death increase after 48 h. Usually, cytotoxicity depends on compounds’ concentration and exposure time [46]. A slight increase in mortality was found for F2 between 24 and 48 h (from 5 to 6%). The increase could be explained by the material’s lumps encapsulating the nauplii, causing their death. However, this percentage (Table 2) indicates that the material’s biocompatibility as it is below 30% [47]. 

Figure 7 shows the InterOss Collagen (F2) material images at 24 and 48 h, indicating the release of material residues and live and dead larvae trapped in them. 

The observed dissolution of F2 seems to be caused by a hydrolytic degradation process that would cause the fragmentation of the material into smaller particles, typical of these fibrous and highly porous materials [23]. 

### 3.3. Cell Viability Assay 

The MTT cytotoxicity assay is considered a susceptible test. It is based on the ability of live cells to convert MTT into Formazan, which accumulates as an insoluble violet product by mitochondrial or endoplasmic reticulum dehydrogenases [48]. To perform the quantification, the formazan is solubilized using some products present in the MTT Kit, and a review is achieved with a microplate reader. Absorbance measurement at OD590 nm is proportional to the number of live HeLa cells [49]. 

The cell viability values have a homogeneous behavior in each material when comparing the values between the first and second tests (Figure 8). F1 (InterOss) and F3 (InterCollagen guiade) xenografts presented cell viability average values above 75%, demonstrating a low cytotoxic behavior against HeLa line cells. 

Different tests on xenografts have shown that these materials are biocompatible, with variable results in cell proliferation compared with the control group or other allogeneic or alloplastic materials [11,50,51,52,53]. The biocompatibility observed for xenografts is consistent with the results obtained for F1 (InterOss) and F3 (InterCollagen Guide), which show very low percentages of cytotoxicity, in contrast to F2 (InterOss Collagen), which present a critical portion of apparent cytotoxic results when compared with the negative control. However, the F2 results depend on the material residues at the bottom of wells that adsorb some formazan crystals and affect the results, giving false-negative results [54]. It was previously reported that components in the microplate wells, if they scatter or absorb light at the same wavelength of the assay, interfere with the final results [55]. For instance, they could affect the absorbance measurements when performing the readings, similar to the test with brine shrimp. 

The statistical analysis of the treatments demonstrated that there were no statistical differences between the formulations and for each assay, indicating the reproducibility and biocompatibility of the formulations. In the paired samples *t*-test analyses, it was observed that there were no statistically significant differences for both trials in the cell viability MTT test for each material, demonstrating that all the samples are biocompatible and with no significant differences compared to the negative control (Table 3). Those results also demonstrate the reproducibility of the test and its homogeneity.

### 3.4. Cell Growth Assay 

When placing the samples of the three formulations in culture wells with bone cells, it was found that the InterCollagen guide samples (F3) floated in the wells, and we decided we do not continue the experiment with these samples. On the other hand, the InterOss (F1) and InterOss Collagen samples (F2) were distributed at the bottom of the well. 

After six days, the control wells presented a confluence greater than 75% (Figure 9A,B). When examining the wells of the F1 (InterOss) and F2 (InterOss Collagen) formulations, the entire presence of cells throughout the surface was observed, indicating that the materials did not affect the growth and development of the cells after their placement, despite the presence of residues from the InterOss Collagen material (Figure 9C,D). 

The way that InterOss (F1) is distributed over the surface and the residues released from the InterOss Collagen material (F2) made it difficult to observe under the microscope; but, the presence of cells adhered to different parts of the surface indicates that cell growth occurred and that the two materials were compatible.

### 3.5. In Vivo Biocompatibility Assay 

Once the models that completed the implantation periods (30, 60, and 90 days) were euthanized, the macroscopic appearance of the implantation zones was verified. Figure 10 corresponds to the macroscopic inspection of the biomodels. At 60 days, there was complete hair recovery, according to the images. When performing hair removal by trichotomy, complete healing of the implanted area was observed. On the other hand, when reviewing the internal surface of the skin, the sites where the materials F1 and F2 were embedded, it was shown that the materials were not very visible. Additionally, necrosis sites were absent. 

#### 3.5.1. Results of Subdermal Implantation of F1 (InterOss)

Little evidence of degradation was found in F1 in the three observation periods. However, at 60 and 90 days of implantations, it is possible to observe slight changes in the surfaces of the particles and an increase in cellular activity, indicating the beginning of the phagocytic degradation process and subsequent material’s reabsorption. 

Figure 11 shows in the implantation area after 30 days (Figure 11A), particles surrounded by a capsule of collagen I fibers (Figure 11A,B), and cellular activity around the particles after 60 and 90 days (Figure 11D–F). The implantation area and the particles were surrounded by a structure composed of collagen I fibers, with a slight inflammatory infiltrate. Some multinucleated cells (Figure 11E) and large blood vessels (Figure 11F) were also identified. 

#### 3.5.2. Results of Subdermal Implantation of F2 (InterOss Collagen) 

Thirty days after implantation of F2, it was observed that the implantation area is surrounded by a fibrous capsule similar to that described for F1 (Figure 12A,B) with some evidence of degradation (Figure 12C). After 60 days, the material underwent a disintegration process compatible with a hydrolytic degradation that produced the separation of the apatite particles from the network of collagen fibers where they were immersed (Figure 12D,E). Large blood vessels were observed in all the periods evaluated, which play a critical role in the tissue repair process and increase significantly in the proliferative phase when necessary to guarantee adequate blood flow [56]. Figure 12F shows a large blood vessel in the implantation area at 90 days with fragments of collagen material and bovine apatite. 

#### 3.5.3. Results of Subdermal Implantation of F3 (InterCollagen Guide) 

The material’s behavior was similar to that of the other two materials. The F3 membrane fragments were encapsulated by a network of collagen fibers with the presence of cells and blood vessels. Figure 13 shows the results of the implantation of the biomaterial. 

Figure 13A–C correspond to the 30 days. Little evidence of degradation is observed, and the film fragments are surrounded by a fibrous capsule of collagen I with the presence of inflammatory cells. At 60 days of implantation, complete pieces and some partially degraded membranes are observed (Figure 13D,E). 

Small fragments are observed at higher magnification (100×) in the areas undergoing resorption (Figure 13F). Finally, 90 days after implantation, there are still some fragments of the material immersed in fibrous tissue and the process of reabsorption by cellular activity (Figure 13G,H). Several blood vessels are present in this fibrous tissue (Figure 13I).

## 4. Conclusions

Thermogravimetric analysis (TGA) demonstrated thermal stability and the presence of a low amount of organic material from the residual weight for the three formulations. From the Fourier Transform Infrared Spectroscopy (FT-IR), we evidenced the presence of hydroxyl, orthophosphate, carbonate groups of the xenografts, and amide groups of collagen. Analysis by differential scanning calorimetry showed the presence of endothermic peaks (73, 82, and 226 °C for F1 and F2, respectively) typical of the dehydration of the xenografts and collagen membrane (F3). In comparison, DTGA evidenced the beginning of protein decomposition at 334 °C. 

The three materials tested showed in vitro and in vivo biocompatibility. The in vitro tests showed that the three materials (F1, F2, and F3) are biocompatible since they allowed the *Artemia salina* growth, bone cells, and MTT cell viability of HeLa cells. It was also observed that formulation F2 separates tiny fragments in contact with the aqueous medium (solution of seawater and culture medium). The observation of the disintegration presented by formulation F2 seems to influence the degradation process in vivo by facilitating the action of phagocytic cells, as evidenced in the histological results. 

The healing process of the tissues after the InterOss (F1) and InterOss Collagen (F2) samples was carried out as a reaction to a foreign body with encapsulation of the materials from a collagen network I fibers. The healing process also had a slight inflammatory infiltrate and large blood vessels, according to the results presented and like what has been described as the typical healing mechanism for biomaterials. However, the InterCollagen Guide (F3) showed almost complete reabsorption of sample implanted without encouraging a foreign body reaction and with little inflammatory infiltrate. 

The three materials presented biocompatibility in vitro and in vivo and stability that would guarantee their presence in contact with tissues for more than three months. F1 (InterOss) and F3 (InterCollagen Guide) showed strength in all implantation periods. At the same time, the F2 material (InterOss Collagen) presented an initial degradation that caused the separation of the apatite particles from the collagen network, observing collagen fragments and apatite particles simultaneously in the 60 days. On the other hand, formulation F3 shows a more resorption process when compared to the other two, but small particles are still observed at three months.

## Figures and Tables

**Figure 1 polymers-14-02672-f001:**
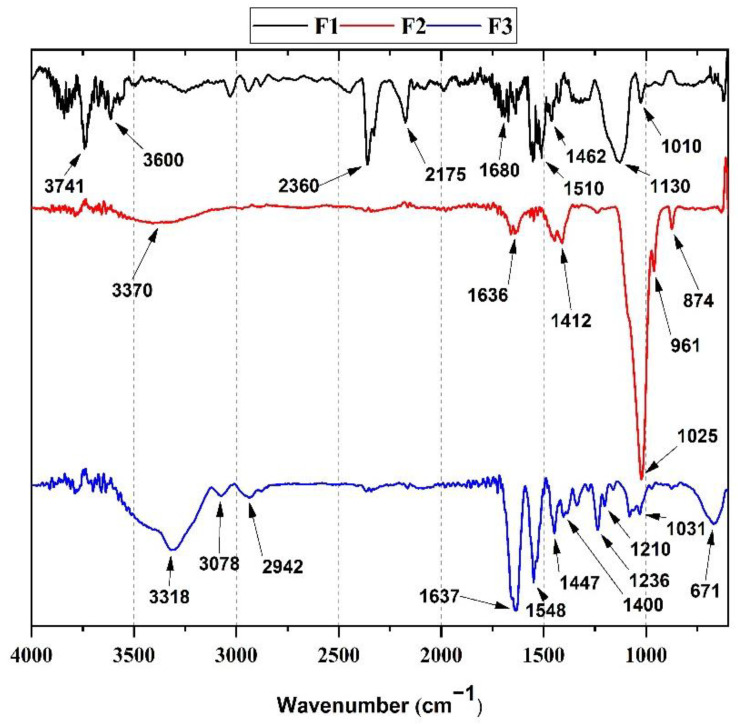
FT-IR spectrum of F1, F2, and F3. According to Table 1, F1: InterOss. F2: InterOss Collagen. F3: InterCollagen Guide.

**Figure 2 polymers-14-02672-f002:**
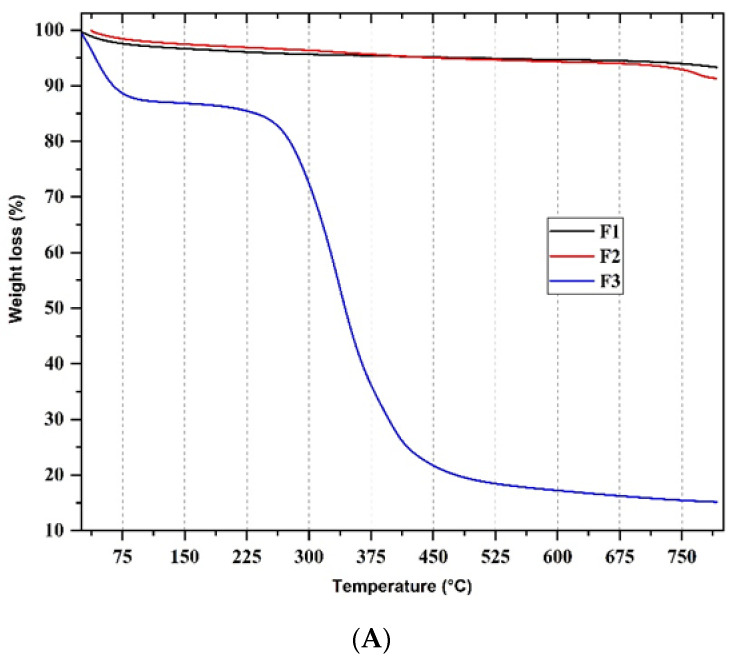
TGA and DTGA analysis of F1, F2, and F3. According to Table 1, F1: InterOss. F2: InterOss Collagen. F3: InterCollagen Guide. (**A**) Thermogravimetric analysis (TGA). (**B**) Derivative TGA.

**Figure 3 polymers-14-02672-f003:**
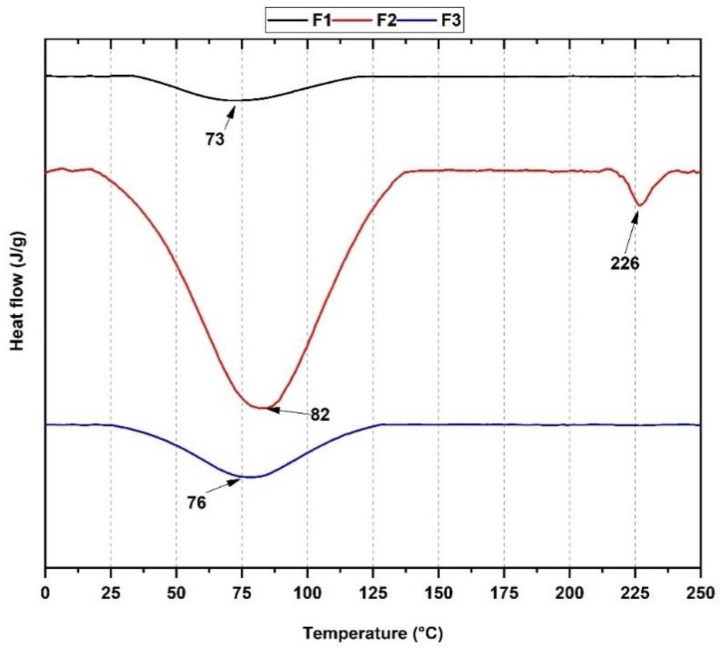
DSC thermograms for F1, F2, and F3. According to Table 1, F1: InterOss. F2: InterOss Collagen. F3: InterCollagen Guide.

**Figure 4 polymers-14-02672-f004:**
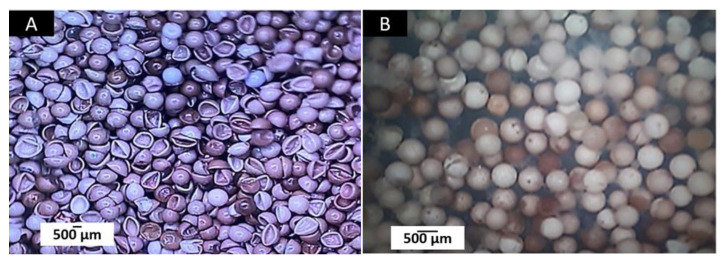
Cysts and larvae of *Artemia salina*. (**A**) Dry cysts at 3×. (**B**) Hydrated cysts at 24 h at 3×. Images were taken with a stereoscopic microscope.

**Figure 5 polymers-14-02672-f005:**
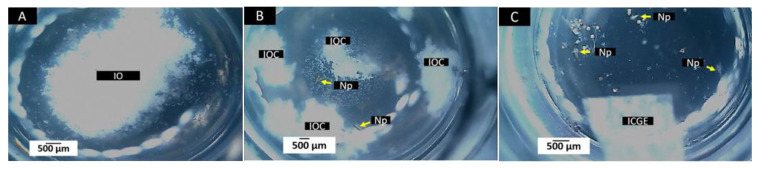
Samples of the materials were incubated for 24 h in a seawater solution with larvae of *A. salina*. (**A**) Sample F1. IO = Material F1. (**B**) Material F2. 9IOC = Material F2 NP = Nauplius. (**C**) Material F3. ICGE= Material F3. NP= Nauplius. Images at 3×. Images were obtained with a stereoscopic microscope.

**Figure 6 polymers-14-02672-f006:**
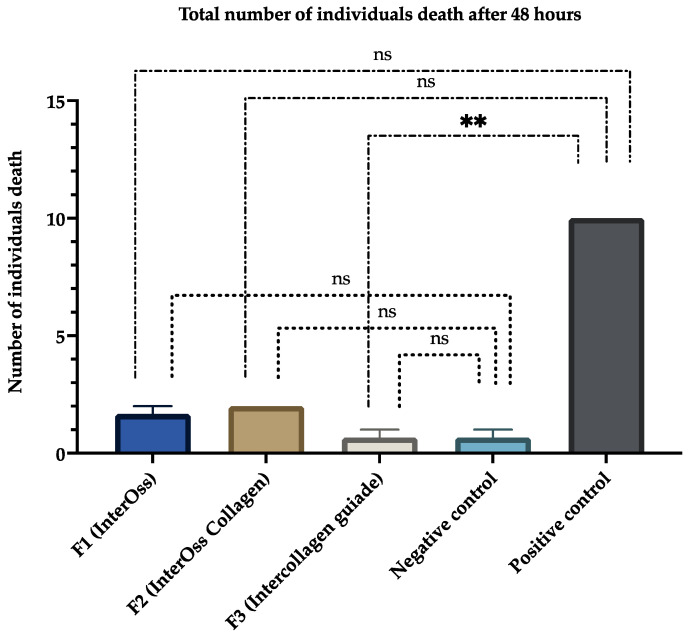
Number of individuals death for *A. salina* at after 48 h. (ns: non-significant differences between treatments and assays; ** *p* < 0.01).

**Figure 7 polymers-14-02672-f007:**
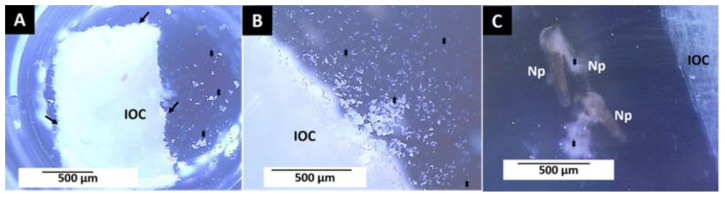
Material F2 (InterOss Collagen) at 24 and 48 h in a seawater solution with *Artemia salina*. (**A**) Material at 24 h. Image at 1×. (**B**) Material at 48 h. Image at 3×. (**C**) Material at 48 h. Image at 4×. IOC: Material. Np: Nauplii. Stars: Detached material. Stereoscopic microscope technique. IOC: Material. Black arrow: areas of material degradation. Red arrow: Fragments of the detached material.

**Figure 8 polymers-14-02672-f008:**
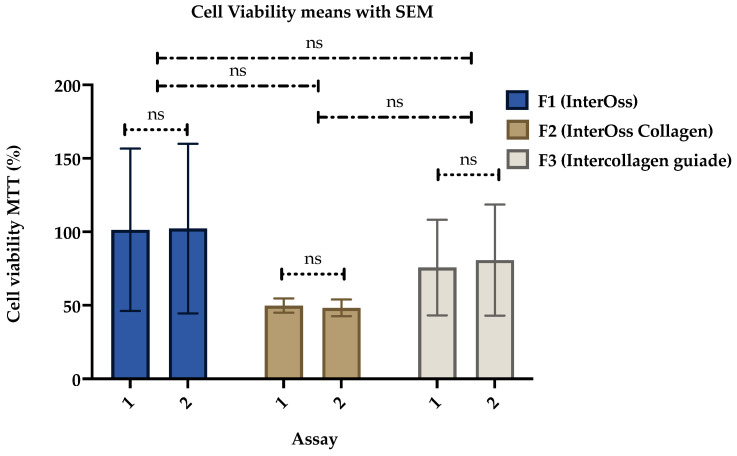
Mean and standard error of cell viability MTT for the xenografts.

**Figure 9 polymers-14-02672-f009:**
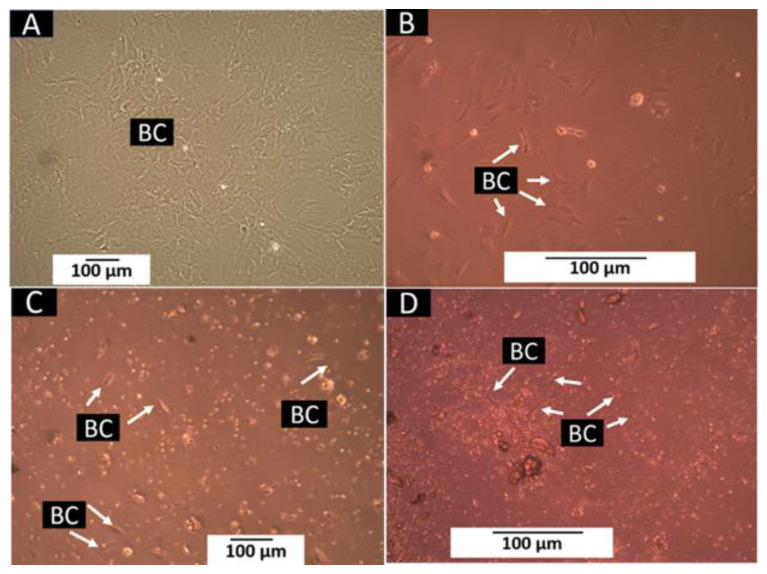
Xenograft samples were cultured with bone cells for six days. (**A**) Growth control well at 10×. (**B**) Growth control well at 20×. (**C**) Well, with F1 at 10×. (**D**) Well, with F2 at 10×. BC: bone cells.

**Figure 10 polymers-14-02672-f010:**
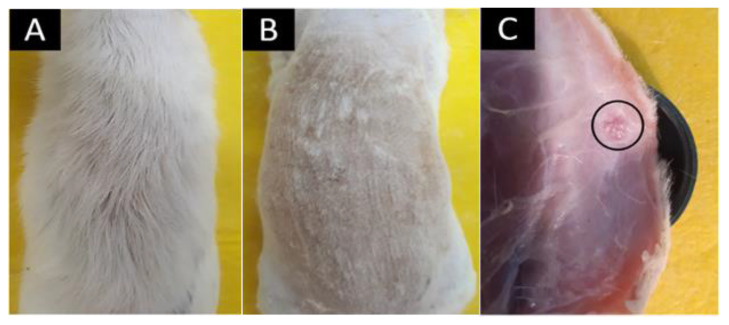
Macroscopic image of the implanted area. (**A**) Dorsal area with hair. (**B**) Dorsal area with a trichotomy. (**C**) Internal surface of the dorsal area. The circle corresponds to one of the sites where the materials were implanted.

**Figure 11 polymers-14-02672-f011:**
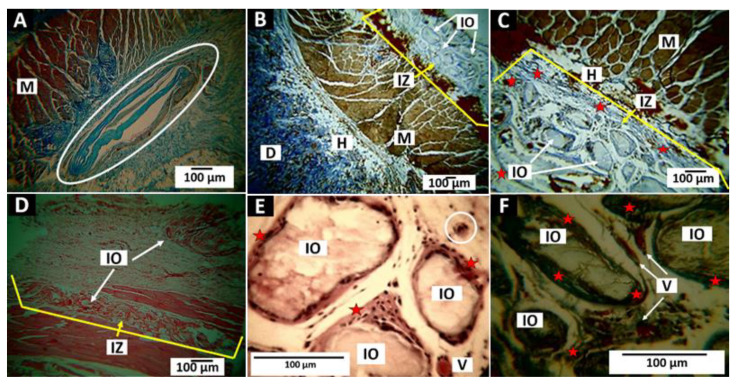
Subdermal implantation of F1 (InterOss). (**A**) image at 10×. MT technique. (**B**) Image at 4×. MT technique. (**C**) image at 10×. MT technique. (**D**) Image at 4×. H-E technique. A, B and C: MT technique. (**D**) Hematoxylin—Eosin technique. (**E**) 40× image, H-E technique. (**F**) Image at 40×. MT technique. Oval: Capsule in implantation area. D: Dermis. H: Hypodermis. M: Muscle. IZ: Implantation zone. IO: InterOss. V: blood vessel. Stars: fibrous capsule. Circle: Multinucleated cells.

**Figure 12 polymers-14-02672-f012:**
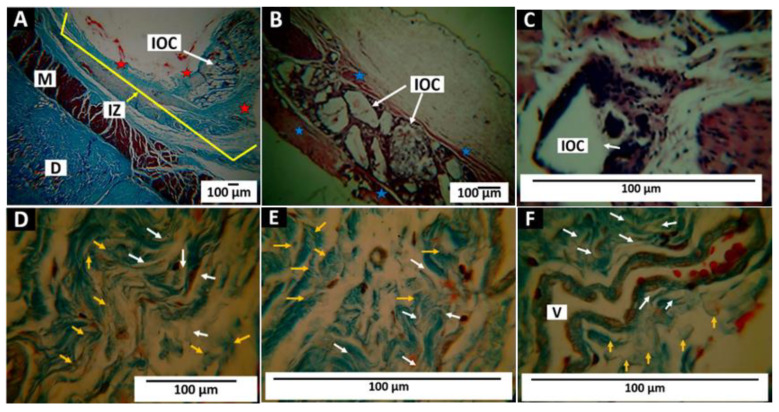
Subdermal implantation of F2 (InterOss Collagen). (**A**) Image at 4×. TM technique. (**B**) Image at 10×. H-E technique. (**C**) Image at 100×. H-E technique. (**D**) Image at 100×. MT technique. (**E**) 100×. TM technique. (**F**) 100×. MT technique. D: Dermis. M: Muscle. IZ: Implantation zone. IOC: InterOss Collagen. Stars: fibrous capsule. White arrow: Collagen fiber fragments of the material. Yellow arrow: Bovine apatite fragments of the material. V: blood vessel.

**Figure 13 polymers-14-02672-f013:**
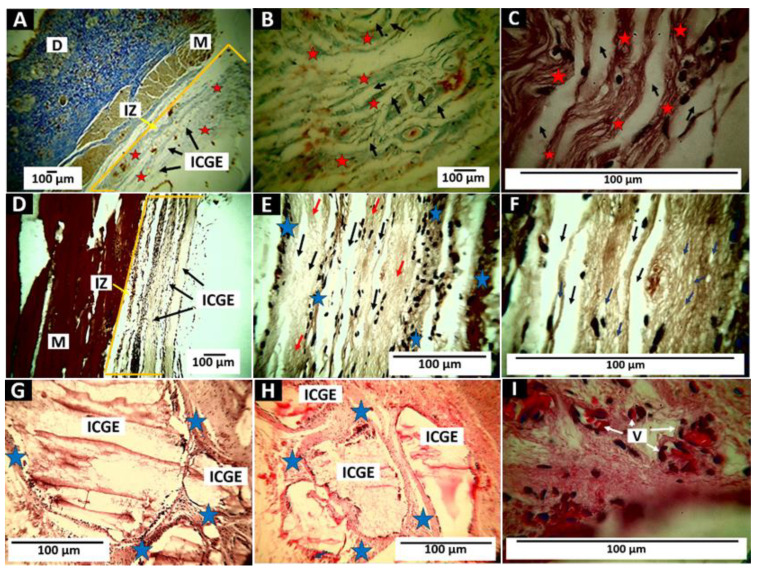
Subdermal implantation of F3 (InterCollagen Guide). (**A**) Image at 4×. MT technique. (**B**) 10×. MT technique. (**C**) 100×. H-E technique. (**D**) 10×. H-E technique. (**E**) 40×. H-E technique. (**F**) 100×. H-E technique. (**G**) 40× H-E technique. (**H**) 40×. H-E technique. (**I**) 100×. H-E technique. D: Dermis. M: Muscle. IZ: Implantation zone. IOC: InterOss Collagen. Black arrows: ICG fragments. Red arrows: Area of material’s reabsorption. Blue arrows: pieces of the material in the resorption zone. Stars: a fibrous capsule that surrounds the material. V: blood vessels. subfigure (**A**–**C**): Material implanted at 30 days. (**D**–**F**): Material implanted at 60 days. (**G**–**I**) Implanted at 90 days.

**Table 1 polymers-14-02672-t001:** The formulation composition of the xenografts used in this research.

Samples	Formulation	Composition
InterOss	F1	Xenograft of 100 wt.% bovine bone particles
InterOss Collagen	F2	Xenograft of 90 wt.% of bovine particles and 10 wt.% of porcine collagen fibers
InterCollagen Guide	F3	The membrane of porcine collagen

**Table 2 polymers-14-02672-t002:** Dead larvae (n) and mortality percentage (%) of *A. salina* per sample at 24 and 48 h.

Sample	24 h n (%)	48 h n (%)	Total n (%)
F1	1 (3.3)	4 (13.8)	5 (17.1)
F2	1 (3.3)	5 (17.2)	6 (20.5)
F3	0 (0)	2 (6.7)	2 (6.7)
Positive control	18 (60)	12 (100)	30 (100)
Negative control	0 (0)	2 (6.7)	2 (6.7)

**Table 3 polymers-14-02672-t003:** Paired samples *t*-test between experimental materials.

	1st Assay			2nd Assay		
Paired Samples *t*-Test	Control (-)	F1	F2	Control (-)	F1	F2
**F1**	0.1808	-	-	0.1808	-	-
**F2**	0.1049	0.3661	-	0.0691	0.2843	-
**F3**	0.1049	0.3661	0.5000	0.1524	0.4546	0.3242

## Data Availability

Data from the experiments are available upon request from the corresponding author.

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
