# Peer review of "Biocompatibility Assessment of Two Commercial Bone Xenografts by In Vitro and In Vivo Methods"

_polymers, 2022, doi:10.3390/polym14132672_

Round 1

Reviewer 1 Report

The authors report a study in which they compared the biological response of two commercial xenografts (InterOss® and InterOss® Collagen) and a commercial porcine collagen membrane (InterCollagen® Guide). Various experimental techniques were used, which allowed the interaction of the materials with bone to be studied at the level of essential chemistry.   Results showed that the materials interacted and produced distinct chemical changes, demonstrating the bioactivity  of these materials.  In vitro biocompatibility was also determined using both Artemia salina and MTT cell viability testing. Finally biocompatibility was studied using Wistar rat models with implanted materials allowed to reside in situ for different time periods (30, 60, and 90 days). These biological results confirmed the high biocompatibility of these materials.

Overall this is a very useful study, and merits publication without modification.

Author Response

Reviewer 1

The authors report a study in which they compared the biological response of two commercial xenografts (InterOss® and InterOss® Collagen) and a commercial porcine collagen membrane (InterCollagen® Guide). Various experimental techniques were used, which allowed the interaction of the materials with bone to be studied at the level of essential chemistry.   Results showed that the materials interacted and produced distinct chemical changes, demonstrating the bioactivity  of these materials. In vitro biocompatibility was also determined using both Artemia salina and MTT cell viability testing. Finally biocompatibility was studied using Wistar rat models with implanted materials allowed to reside in situ for different time periods (30, 60, and 90 days). These biological results confirmed the high biocompatibility of these materials.

Overall this is a very useful study and merits publication without modification.

R// We are deeply thankful for the positive comment and recommendation from the reviewer.

Reviewer 2 Report

The manuscript “Biocompatibility assessment of two commercial bone xenografts by in vitro and in vivo methods” addresses an interesting topic and clearly describes the experimental methods and results.

Moreover, the paper also uses a good English. Therefore, it may be recommended for publication after minor revision:

1)     I suggest to the authors to acquire again the FTIR spectra, to check the reported peaks according to the inserted references and to explain in results by mentioning the precise peak and not an entire band.

2)     I advise the authors to review the statistical analysis for all the bioligical characterization and to insert it where it is missing

3)     In the introduction part it is recommended to add some more discussion. The following publication is recommended to fulfill this section:              CATAURO, Michelina, et al. Antibacterial Properties of Sol–Gel Biomaterials with Different Percentages of PEG or PCL. In: Macromolecular Symposia. 2020. p. 1900056

Author Response

Reviewer 2

Comments and Suggestions for Authors

The manuscript "Biocompatibility assessment of two commercial bone xenografts by in vitro and in vivo methods" addresses an interesting topic and clearly describes the experimental methods and results.

Moreover, the paper also uses a good English. Therefore, it may be recommended for publication after minor revision:

  • I suggest to the authors acquire again the FTIR spectra, to check the reported peaks according to the inserted references, and to explain in results by mentioning the precise peak and not an entire band.

R// We appreciate the suggestion from the reviewer. However, acquiring a new FT-IR spectrum was impossible since we ran out of the sample. Besides, it is demanding that all the bands match the inserted references since samples correspond to different batches of fabrication. On the other hand, FT-IR spectroscopy is more appropriate to indicate bands than peaks. Despite that, we did several corrections to better understand the bands and peaks and highlighted them in the figure and between lines 261-287.

  • I advise the authors to review the statistical analysis for all the biological characterization and to insert it where it is missing

 R// We appreciate the valuable suggestion from the reviewer. After checking the biological characterization section, some information was corrected. For example, between lines 390-393, the text was corrected to: "Also, the results demonstrated that there were no significant differences between treatments and treatments with the negative control, but there was a significant difference (p < 0.01) between treatment F3 with the positive control, indicating the high biocompatibility of the materials (Figure 6). "

We did this correction to understand better the significant differences between the tests with the positive control but not the negative control and between the treatments.

Also, we checked the MTT results and added information to Figure 8, where we established significant differences between the treatments, which might be due to the F2 adhesion problem at the bottom of the well. However, we had the text (lines 436 – 442) explain that:

 "The statistical analysis of the treatments demonstrated that there were no statistical differences between the formulations and for each assay, indicating the reproducibility and biocompatibility of the formulations. In the paired samples t-test analyses, it was observed that there were no statistically significant differences for both trials in the cell viability MTT test for each material, demonstrating that all the samples are biocompatible and with no significant differences compared to the negative control (Table 3). Those results also demonstrated the reproducibility of the test and the homogeneity."

On the other hand, the growth cell test and the subdermal implantations do not have statistical analysis since they are qualitative experiments, and we cannot measure statistical differences.

3)     In the introduction part it is recommended to add some more discussion. The following publication is recommended to fulfill this section: CATAURO, Michelina, et al. Antibacterial Properties of Sol–Gel Biomaterials with Different Percentages of PEG or PCL. In: Macromolecular Symposia. 2020. p. 1900056

R// We appreciate the suggestion. The correction was done between lines 52-62 of the introduction section, including five new updated references.

"Alloplastic materials have been developed and are proposed as potential bone substitutes. Their outstanding versatility allows them to be processed with different techniques such as 3D printing to obtain porous structures that mimic cancellous bone [5] or resemble the fibrillar portion of the extracellular matrix through nanofibers [6].

Synthetic materials also have the advantage of the possibility of incorporating components with anti-inflammation and antibiotic effects to reduce the possibility of complications [7] or biological factors such as bone morphogenic proteins (BMP-9) to achieve an osteoinductive effect [8].

 Several bone substitutes currently use synthetic materials combined with natural products modified in the laboratory, such as collagen, to achieve biomimetic scaffolds that resemble the extracellular bone matrix [9]."

Reviewer 3 Report

The research manuscript entitled “Biocompatibility assessment of two commercial bone xenografts by in vitro and in vivo methods” by Valencia-Llano et al. for Polymer journal. The study has be performed logically and details experimental results confirmed the outcome of bone regeneration. I will suggest for minor revision.

Following concerns should be thoroughly addressed before consider for acceptance:

1.     In Abstract, line no.18-20; “Fourier Transform Infrared Spectroscopy (FT-IR)………..(FT-IR), should be corrected.

2.     Though, authors have provided detailed introduction, however, it would be great if authors can add important recent papers which highlights the bone regeneration using different scaffolds such as nanofibers, 3D printed, and hydrogels and also provide brief details about Artemia salina. Following Refs could be useful : Acta Biomaterialia 108 (2020) 97–110, Acta Biomaterialia 10 (2014) 1238–1250, Materials & Design 210 (2021) 110049.

3.     Page 3, line# 121, “5 wt.% CO2” should be changed to “5% CO2”.

4.     In FT-IR data (Figure 1), “Wavenumber (cm)-1” should be replaced by “Wavenumber (cm-1)”

5.     In Figure 2 caption, Please provide (A) and (B) legends in detail.

6.     In Figure 4 (A) and (B), please provide the scale bar. Please provide the scale in figures 5 too.

7.     In Figure 8, why standard deviation is very high? Please explain.

8.     In Figure 9, Please provide the scale bar.

Author Response

Reviewer 3

The research manuscript entitled "Biocompatibility assessment of two commercial bone xenografts by in vitro and in vivo methods" by Valencia-Llano et al. for Polymer journal. The study has be performed logically and details experimental results confirmed the outcome of bone regeneration. I will suggest for minor revision.

Following concerns should be thoroughly addressed before consider for acceptance:

  1. In Abstract, line no.18-20; "Fourier Transform Infrared Spectroscopy (FT-IR)………..(FT-IR), should be corrected.

R// We appreciate the suggestion. The correction was done accordingly between lines 20-22:

"Fourier Transform Infrared Spectroscopy (FT-IR) analysis evidenced the presence of hydroxyl, orthophosphate, and carbonate groups of the xenografts and amide groups of collagen."

  1. Though, authors have provided detailed introduction, however, it would be great if authors can add important recent papers which highlights the bone regeneration using different scaffolds such as nanofibers, 3D printed, and hydrogels and also provide brief details about Artemia salina. Following Refs could be useful : Acta Biomaterialia 108 (2020) 97–110, Acta Biomaterialia 10 (2014) 1238–1250, Materials & Design 210 (2021) 110049.

R// We appreciate the suggestion. We added new information between lines 52-62 of the introduction section, including five new updated references.

"Alloplastic materials have been developed and are proposed as potential bone substitutes. Their outstanding versatility allows them to be processed with different techniques such as 3D printing to obtain porous structures that mimic cancellous bone [5] or resemble the fibrillar portion of the extracellular matrix through nanofibers [6].

Synthetic materials also possess the advantageous possibility of incorporating components with anti-inflammation and antibiotic effects to reduce the possibility of complications [7] or biological factors such as bone morphogenic proteins (BMP-9) to achieve an osteoinductive effect [8].

 Several bone substitutes currently use synthetic materials combined with natural products modified in the laboratory, such as collagen, to achieve biomimetic scaffolds that resemble the extracellular bone matrix [9]."

We also added more information about the Artemia salina test between lines 340-344:

"The A. salina test involves exposing 24-hour hatched larvae (instars II-III) to different materials or concentrations of the same material to determine the material's biocompatibility or the lethal dose 50 (LC50) of a specific concentration of the material. Larvae are expected to be affected by the material if it is not compatible. This toxicity test is considered reliable and replicable [36]."

  1. Page 3, line# 121, "5 wt.% CO2" should be changed to "5% CO2". (Yo)

R// We appreciate the suggestion. The correction was done accordingly.

  1. In FT-IR data (Figure 1), "Wavenumber (cm)-1" should be replaced by "Wavenumber (cm-1)"

R// We appreciate the suggestion. The correction was done accordingly.

  1. In Figure 2 caption, please provide (A) and (B) legends in detail.

R// We appreciate the suggestion. We removed the information from the figure and added it to the caption for clarity.

  1. In Figure 4 (A) and (B), please provide the scale bar. Please provide the scale in figures 5 too.

R// We appreciate the suggestion. The correction was done accordingly in Figures 4 and 5.

  1. In Figure 8, why standard deviation is very high? Please explain.

R// We appreciate the suggestion. For the three materials, three replicates and two different readings were made. The results were very similar when comparing the two readings, but one of the three cells presented higher absorbance values ​​in all three replicates than the other. The cause could be the difficulty of altogether removing the material's residue remaining adhered to the bottom of the well after washing when the treatment with the MTT is going to be carried out, which considerably altered the absorbance value.

  1. In Figure 9, Please provide the scale bar.

R// We appreciate the suggestion. The correction was done accordingly in Figures 7 and 9.
